# The Function of NK Cells in Tumor Metastasis and NK Cell-Based Immunotherapy

**DOI:** 10.3390/cancers15082323

**Published:** 2023-04-16

**Authors:** Yanlin Yu

**Affiliations:** Laboratory of Cancer Biology and Genetics, Center for Cancer Research, National Cancer Institute, National Institutes of Health, Bethesda, MD 20892, USA; yuy@mail.nih.gov; Tel.: +1-240-760-6812; Fax: +1-301-480-4662

**Keywords:** NK cell function, metastasis, NK cell-based immunotherapy, metastatic tumor immune escape, immunosuppression, NK cell activation, CAR-NK cell

## Abstract

**Simple Summary:**

Innate immune natural killer (NK) cells are capable of killing metastatic cancer cells without activation by antigen-presenting cells beforehand. The cytotoxic effects and immune regulation of NK cells are precisely controlled by an energetic balance of signals produced by a group of activating and inhibitory receptors expressed on NK cells, while metastatic tumor cells with multiple strategies escape immune cells attack. This review focuses on the critical function of NK cells in metastasis and recently developed objectively effective NK cell-based immunotherapies.

**Abstract:**

Metastatic tumors cause the most deaths in cancer patients. Treating metastasis remains the primary goal of current cancer research. Although the immune system prevents and kills the tumor cells, the function of the immune system in metastatic cancer has been unappreciated for decades because tumors are able to develop complex signaling pathways to suppress immune responses, leading them to escape detection and elimination. Studies showed NK cell-based therapies have many advantages and promise for fighting metastatic cancers. We here review the function of the immune system in tumor progression, specifically focusing on the ability of NK cells in antimetastasis, how metastatic tumors escape the NK cell attack, as well as the recent development of effective antimetastatic immunotherapies.

## 1. Introduction of the General Immune System

The immune system, composed of a complex of organs, immune cells, and molecules in the body, not only defends against foreign pathogens but also eliminates potentially deleterious mutation-bearing cells to protect and maintain normal body functions (Figure 1). Immune cells originate from stem cells in immune organs and become a variety of cell types, including neutrophils, eosinophils, basophils, mast cells, monocytes, macrophages, dendritic cells, natural killer cells (NK cells), B lymphocytes (B cells), and T lymphocytes (T cells). Physiologically, immune cells maintain a homeostatic balance between activating and inhibitory immunity for an adequate immune reaction against pathogens and diseases, including cancer [1]. The immune reaction contains two primary types: innate and adaptive immunity. Innate immunity is predominantly nonspecific, and defense reactions are rapidly generated within foreign antigen encounters. The NK and dendritic cells, as well as macrophage, neutrophils, basophils, and eosinophils, are the primary cell types in innate immunity [2]. Adaptive immunity is based on the clonal selection of T cells and B cells with receptors recognizing particular ‘non-self’ antigens represented by major histocompatibility complex (MHC) molecules on antigen-presenting cells. It is the second line of protection and needs a few days to become activated. After activation and expansion, T cells become cytotoxicity T cells (CD8+ T cell) to recognize and kill the target cells expressing the same antigen through phagocytosis and antigen-specific cytotoxicity [3], while B cells produce the antibody for antigen and are primarily demanded in antibody-mediated immunity [4] (Figure 1B).

Cancer originates from cells harboring genetic mutations [5], which can express the mutant proteins as neoantigens. Neoantigen peptides are presented to immune cells through the MHC class I (MHC-I) molecules on the surface of the cancer cell, distinguishing them from their normal counterparts (nonself) [6]. This presentation leads to the activation of T cells to become specific cytotoxicity CD8+ T cells, which can kill those tumor cells. NK cells can detect and rapidly destroy tumor cells with downregulated MHC-I expression, terming ‘missing self’ recognition [7,8]. Hereafter, NK cells quickly produce cytokines and secret the interferon-γ (IFNγ), tumor necrosis factor-alpha (TNFα), granulocyte-macrophage colony-stimulating factor (GM-CSF), and chemokines for activating other immune cells such as T and B cells to boost adaptive immunity [9]. Moreover, NK cells can be provoked by antibody opsonization for mediating antibody-dependent cellular cytotoxicity (ADCC) [10]. To trigger and develop the entire immune response requires the interaction of tumor cells and immune cells, which needs three signals: (1) first signal—an antigen expressed in cancer cells; (2) second signal—stimulatory molecules in both cancer and immune cells, and (3) third signal—cytokine signaling in immune cells [11,12]. However, cancers can develop multiple strategies to escape immune cell attack [13]. Immune escape is usually associated with loss of stimulating molecules that includes the downregulation of classical HLA molecules (missing self-hypothesis), loss of stimulatory cytokines, and/or gain of suppressing molecules such as expression of nonclassical HLA-G, functional Th2-type activity shift (e.g., decrease in IFNγ and/or increase in TGFb, IL6, and IL10) and elevation of the Fas ligand on cancer cells [14,15,16]. The interactivity of the immune cells and cancer cells can determine the fate of tumors. This can be described as three “E’s”, which are eradication, equilibrium, and escape (Figure 2) [17,18]. This process, called cancer immunoediting, has built a theoretical hypothesis for comprehending our immune protection and sculpting cancer immunogenicity on tumor progression. In early cancer development, tumor cells can be eliminated by the immune system via the responses of innate and adaptive immunity in the eradication phase. However, tumor cells can manage the immune pressure to survive and then may enter the equilibrium phase. Escape is the phase of the tumor progression, where tumors are sculpted immunologically with the ability to inhibit the immune cells attach and establish an immunosuppressive tumor microenvironment (TME), then become clinically relevant as a mass and/or metastasis [13,17,18,19,20] (Figure 2). Boosting immunity and inhibiting immunosuppressive TME are principal strategies for anticancer immune therapy. One of the breakthroughs in antitumor immunotherapy focuses on utilizing adaptive immunity, achieving primary success through inhibiting immune checkpoints [21] or using CAR-T cells to enhance antitumor CD8+ T cell responses [22,23]. Emerging evidence indicates that tumors can develop various strategies to evade CD8+ T cell recognition, and side effects remain in T cell therapies [24]. Although T cells can eliminate most cells within a primary tumor, a fraction can still escape antitumor immunity and survive, ultimately leading to metastatic disease and death of the patient. However, NK cells can preferentially attack these tumors [25]. NK cells have built-in safety characteristics to avoid autoimmunity and have the ability to maintain immune homeostasis to prevent autoimmune disease [26]. Moreover, a subgroup of NK cells can present memory-like recall responses, playing the role of adaptive immunity [27,28]. Thus, these unique features of NK cells make them attractive targets for immunotherapy that can provoke their potent antitumor mechanisms (Table 1). Consequently, a new field of NK cells in metastasis has highlighted the important role of NK cells in immunosurveillance against metastasis [29]. This review focuses on the critical functions of NK cells in metastasis and the recent development of novel and objectively effective cancer immunotherapies.

## 2. NK Cell Biology and Function

NK cells represent 5–15% of the lymphocytes in human peripheral blood, with the unique ability to rapidly eliminate infected, mutated cells without preactivation [30,31,32]. NK cells originate from CD34+ hematopoietic stem cells in the BM, migrate to blood and other organs, and infiltrate cancer tissue [31,33]. Based on the surface marker profiles, NK cells exhibit different populations with functional heterogeneity at different maturation stages and anatomical locations [34]. Distinguished from myeloid cells, NK cells are defined as CD3^−^CD56^+^ in peripheral blood [35]. Two major subtypes of NK cells are differentiated by levels of marker CD56 (neural cell adhesion molecule, NCAM) [36]. A total of 5–10% of blood NK cells are immature and express high levels of CD56 named CD56 bright NK cells, while most (90–95%) have downregulation of CD56 and convert into the matured CD56dim subgroup and also begin to express CD16 (FcrRIIIa) [37,38]. NK cells play roles in cytotoxic effects and immune regulation, two significant functions in innate immunity [39]. Without prestimulation, NK cells are able to detect and kill transformed cells by secreting perforin and granzymes [40]. Activated NK cells can also deliver the death ligands of TNFa, FasL, and apoptosis-inducing ligand (TRAIL), starting the apoptotic pathway [41]. In immune regulation, NK cells release various cytokines and chemokines such as IFNγ, IL10, CCL3, CCL4, and CCL5, connecting innate immunity to adaptive immunity [42]. NK cells express diverse activating and inhibitory receptors that transfer positive and negative signals for precisely regulating NK cell activity (Figure 3). The quantity of activating and inhibitory receptor signals contributes to the fate of NK cells’ target cells [43]. NKG2D and DNAM-1 are important receptors for activating NK cells. NKG2D can bind to the ligands of NKG2D (NKG2D), such as MICA/B [44] and UL16-binding proteins (ULBPs) [45,46], extending the signals via the adaptor protein DAP10, thereby activating the PI3K singling pathway to trigger cytotoxicity [47]. DNAM-1 can interact with its ligands, such as CD112 and CD155, in cancer cells [48] (Figure 3). CD16 is another highly efficient activating receptor that matured NK cells express [10]. After binding with the Fc region of antibodies, CD16 induces activation signals and triggers ADCC for lysing antibody-coated cancer cells [10]. Studies showed that several therapeutic antibodies, such as the anti-HER2 antibody trastuzumab not only recognized tumor-specific antigens but also increased NK cell infiltration in tumors to trigger ADCC with a role for NK cells in adaptive immunity [49,50]. Immunoglobulin family receptors, NKp46, NKp30, and NKp44 in NK cells, also play a natural cytotoxicity function when activated by metastatic tumor cells [51,52,53,54,55,56] (Figure 3). The killer immunoglobulin-like receptors (KIRs) [57] are the main NK cell inhibitory receptors that specifically bind to MHC-I molecules. Usually, MHC-I molecules are expressed in the normal cell but are lost in cancer and metastatic cells. By detecting MHC-I molecules, NK cells are inhibited by self-molecules, so they avoid killing normal cells and check “missing self” cancer and metastatic cells using their inhibitory receptors. NKG2A is another inhibitory receptor in NK cells. Upon interacting with nonclassical human leukocyte antigen class I molecule E (HLA-E), NKG2A triggers an inhibitory signal through the immunoreceptor tyrosine-based inhibition motif (ITIM) [58]. Furthermore, the inhibitory leukocyte immunoglobulin-like receptor-1 (LIR-1) could bind to HLA-G and HLA-I to inhibit NK cell activation [59]. Moreover, other inhibitory receptors, TIGIT, CD96, LAG3, PD-1, and Tim-3, express in NK cells for negatively regulating NK cell activation. CD96 and TIGIT can also bind to CD112 and CD155 competitively with DNAM-1 to play checkpoint action for maintaining a balance of activating and inhibitory signals. These receptors also increase as cancer progresses, causing NK cell exhaustion to act as an NK checkpoint, suggesting that these are potential targets in cancer immunotherapy [60] (Figure 3).

## 3. The Function of NK Cells in Tumor Metastasis

NK cells constitutively express lytic machinery that kills aberrant cells independently from preactivation with safety characteristics and rarely elicits autoimmunity (Table 1). Since their identification, NK cells with unique functional features have been suggested to play an important role in controlling tumor growth and metastasis [61,62,63].

### 3.1. NK Cell Function Related to Metastasis

NK cells were first found for their capability to eliminate tumor cells without prior stimulation by antigen-presenting cells [63,64]. Studies have reported that patients with a higher cancer incidence had inadequate peripheral NK cell cytotoxicity responses in various types of cancer [65,66,67,68,69]. Additionally, the patients who suffered NK cell dysfunction had an increased rate of malignancies and metastases [70,71,72,73,74,75,76,77]. Using an experimental metastasis assay, we and others found that antibodies mediated depletion of NK function or the use of an NK cell-deficient host increased metastasis, suggesting that NK cells play an essential role in antimetastasis ([78], our unpublished data). An intravital imaging system directly observed NK cells attack disseminated tumor cells leading to cancer cell death in mouse models [78]. These observations indicated that NK cells act as a killer in tumor progression, especially cancer cells spreading from their original site to the bloodstream [29,62]. Clinically, the patients who suffered various cancers with low amounts of peripheral or infiltrating NK cells at tumor sites have higher numbers of metastatic lesions. [74,79,80,81,82,83,84]. In contrast, the patients who suffered an increased metastatic cancer risk with high levels of NK cell activating receptors (NKAR) have good prognoses [77,85,86,87,88,89]. Moreover, a high level of IFNγ production by circulating NK cells and the presence of NKp30 are associated with a positive prediction for long-term survival in patients with breast cancer, gastrointestinal stromal tumor, or melanoma under treatments [69,79,80,90,91,92]. More recently, a study has shown that NK cells could sustain breast cancer dormancy for controlling breast cancer liver metastasis [93]. These facts suggest the notion that NK cells mediate antimetastatic effects in clinical. The notion has also been strongly linked to cancer treatment and preventing tumor metastasis. For example, after successfully removing the tumor by surgery, many patients still developed distant metastasis later; one of the important reasons is that low NK cell cytotoxicity and less IFNγ secretion link impaired NK cell function directly to increased postoperative metastases [94,95,96]. Thus, the functional restoration of NK cells by agents such as arginine prevents metastases [97,98].

### 3.2. How NK Cells Kill Metastases

It is well appreciated that the amount of positive and negative signals transmitted by NK cell activating or inhibitory receptors determines the fate of the targets [99]. Normal cells typically express low amounts of activating ligands and have higher amounts of MHC-I molecules that interact with inhibitory receptors and transduce more negative signals; therefore, NK cells will not eliminate normal somatic cells. In contrast, in malignant tumor cells, the expression of activating ligands has been increased, and the positive signals will overcome the negative signals, thus ensuring that aberrant cells are destroyed by NK cells [100,101]. The deficiency of NK cell activating receptor NCR1 in a GEM (genetically engineered mouse) model was shown to promote tumor growth [101,102]. Interestingly, when comparing with B16 melanoma growing subcutaneously in either NCR1 heterozygous (het) or homozygous (KO) mice, the tumor size in the two groups had no significant difference. Still, there was a substantial difference in the rate of metastasis between the two groups: NCR1 KO mice appeared to have significantly higher metastatic lesions than NCR1 het mice [102,103]. This indicates that the absence of an activating receptor dramatically affects metastasis development but has no evident effect on the primary tumor, implying that the level of NK cell activating signaling plays a vital role in controlling tumor metastasis. Subsequently, it was found that elevated secretion of IFNγ by NK cells enhanced the level of FN1 in the tumor, resulting in architectural alteration and less aggressive metastasis. In contrast, deficient NCR1 abolished the release of IFNγ [103]. Recent studies have also reported that NK cells are necessary for the selective destruction of circulating single breast tumor cells [104,105]. Extensive studies also support that NK cells control metastasis by activating receptor signaling. For example, NK cells efficiently eliminate metastatic melanoma cells when overexpressing ligands for NKp44, NKp46, and DNAM-1 [74]. In contrast, GEM mice deficient in DNAM-1 [106,107], Tlr3 (regulating NK cell responses to cytokines) [108], Il2rg (ablating NKp46+ NK cells) [109], or T-bet (regulating the differentiation of NK cells) [110,111] are susceptible to metastatic colonization. Interestingly, the metastatic potential could be prevented by transplanting bulk NK cells [110,111,112,113,114,115,116]. Moreover, treating NK cells with activating cytokine IL-15 can restore protection from metastasis in Tbx21-deficient mice [111]. Deletion of a negative regulator Clbl also promotes NK cell-dependent antimetastatic effects in mice [115]. Furthermore, deficient endogenous IL-15 inhibitor of Cish in NK cells (Cish^−/−^NK cells) leads to NK cell hyperactivation; transplanting the Cish^−/−^NK cells into mice could robustly abrogate the metastatic phenotype of highly metastatic B16F10 melanoma cells [117,118]. Decrease in DNAM-1 limits NK cytotoxicity and blocks IFNγ production, while the overexpression of DNAM-1 ligand in tumor cells causes a decrease in NKG2D ligands, indicating that NK cell-induced killing is initiated by DNAM-1 or NKG2D signaling pathway [74,117,119]. Moreover, NK cell depletion with antibodies of NK1.1 or asiago-GM markedly enhances the metastasis [116,120,121,122]. Mice with NK cell activating factors, including IFNγ, perforin 1 (PRF1), or TRAIL deficiency by gene knockout or antibody-caused neutralization, were more susceptible to metastatic incidences following challenges with tumor cell inoculation or with carcinogen-induced tumors [123,124,125]. NK cells also produce and release IFNγ and TNFα to function on macrophages and dendritic cells for enhancing the immune response [126]. Interestingly, using an image tracker system, a recent study functionally visualized that NK cells directly contacted metastatic tumor cells rapidly, leading to its ERK activation and metastatic tumor cell apoptosis [78]. They then confirmed metastatic tumor cell death related to DNAM-1 activation in NK cells [78]. Whereas a more recent study has shown that activation of inhibitory receptor NKG2A/HLA-E signaling promoted the distant metastasis of PDAC; blocking this pathway provokes NK cells and inhibits PDAC metastasis [127]. Whether the NK cells destroy metastatic cancer depends on the amounts of signals from activating and inhibitory receptors in the NK cells. Activating receptors interact with molecules on the surface of cancer cells and ‘turn on’ activation of the NK cell. Inhibitory receptors on NK cells check the signals from its ligands on metastatic cancer cells to block the ability of NK cell-mediated killing. Metastatic tumor cells often lose NK cell inhibitory receptor ligands of MHC-I, which reduce the inhibitory signal and leave them vulnerable to NK cell killing. Activating signals through activating receptors in NK cells promote NK proliferation and stimulate the secretion of cytotoxic granules to release perforin and granzymes, leading to metastatic tumor lysis (Figure 1B and Figure 3).

## 4. Immunosuppression of NK in Tumor Metastasis

The effectiveness of NK cell killing of metastatic tumors depends not only on the immensity of the NK cell response, such as the activation of NK cells, but also on the capacity of cancer cells to evade destruction. Tumor cells can develop a wide range of strategies to elude detection and destruction by NK cells, continue to grow at distant sites, and then form metastases [61]. The process by which tumor cells develop the intrinsic properties to avoid recognition and elimination by NK cells is referred to as “escape” [61]. The efficiency of escape determines the success of disseminated tumor cells in giving rise to distant metastases. Most strategies tumor cells use to evade NK cell killing are described here. First, tumor cells can increase ligands of NK inhibitory receptors such as CD111 [128], PD-L1 [129], HLA-G [130], galectin-9 [131], HMGB-1 [132,133,134], and CEACAM-1 [135], therefore the activation of inhibitory receptor signal pathways [127]. The expression of nonclassical HLA-G is reported in various metastatic cancers, which ties up the inhibitory receptor of LIR-1 to transduce the inhibitory signals to NK cells and inhibit the NK cell proliferation and gene expression (Figure 3) [130]. In fact, cancer cells frequently lose their MHC-1 to escape T cell attack but are vulnerable to NK cell killing, and the metastatic tumor has too many tricks to manage the MHC-1 level for refraining both T cell and NK cell elimination [67], such as genetic and epigenetic modification, and transcriptional and translational regulation [136,137,138]. Second, metastatic cancer cells decrease the ligands of NK cells activating receptors (NKAR) [139,140]. For example, metastatic tumor cells often shed the NKG2D ligand of MICA and MICB proteins [141] through proteolytic proteins such as ADAM10, ADAM17, and MMP14 [142,143], thereby producing soluble variants of ligands, acting as molecular decoys for blocking NK cell activation [144,145]. Indeed, in many different cohorts of cancers, patients with advanced stage tumors often have a high level of soluble NKAR ligands [146,147]. Loss of PVR and nectin 2, a ligand for the NK cell activating receptor DNAM-1, abolished NK cell-mediated destruction of metastatic melanoma B16F10 [78]; the consequence of deletion of DNAM-1 ligand caused due to failure to activation of ERK signal pathway in NK cells [78]. The downregulation of a death receptor FAS in metastatic tumors is another way to turn off NK cells for killing the metastatic cells through the FAS/FASL pathway [148]. Tumor cells can also decrease immunostimulatory factors within the TME. Tumor cells can secrete IL10, CXCL8, and TGFB1, which directly reduce NK cell cytotoxic functions [149,150,151] and/or recruit other immune cells such as Treg cells [152], MDSCs [153], CD11b + Ly6G + neutrophils [122,154], and DC [155], thus inhibiting NK cell functions indirectly. Analysis of the secreted protein profiling of progressive cancers showed little amounts of NK cell-stimulating molecules such as IFN [156] and IL-15 [157]. Tumor cells can rewrite their metabolic programming and release the metabolites that affect the TME to interfere with the antimetastatic functions of NK cells. For example, overexpression of the ectonucleoside triphosphate diphosphohydrolase 1 (ENTPD1) in metastatic tumor cells hydrolyze extracellular ATP into AMP to convert it to adenosine in hypoxia condition caused by HIF-1 [158,159]. Adenosine unleashes powerful immunosuppressive effects on NK cells via adenosine A2a receptor (ADORA2A) signaling [159,160,161,162]. Hypoxia in TME favors metastatic tumor cells for avoiding NK cell elimination by releasing TGFB1 and miRNAs through exosome to target NKG2D [163] and trigger cell autophagy to block tumor cells to GZMB-mediated lysis [164]. In contrast, hyperoxia promotes the destruction of metastases through immune responses of CD8+ CTLs and NK cells [165]. Inhibition of ENTPD1 by small molecular inhibitor polyoxometalate-1 or Entpd1 deletion blocked the metastasis of melanoma and colon cancer cells [166]. In addition, lactate produced by tumors creates a favorable niche for metastasis and modulates the TME preventing NK cell activation [167,168]. Furthermore, tumor-derived stromal inflammation has a condition-dependent effect on controlling metastasis by NK cells [169,170]. Recent studies showed other proteins such as PAEP, pp12, and pp14 upregulated in metastatic tumor cells to hamper NK cell function [171,172]. Last, some reports have shown that metastases exhibited higher stemness features than primary tumors, suggesting that metastatic tumor cells may obtain immune escape abilities from stem cells or dormant cells [173,174] that could shield their proliferation and hide in distant sites for a long time.

## 5. NK Cell-Based Immunotherapy for Metastasis Therapy

Metastatic tumors are challenging to treat and are the principal driver of cancer-related death [175]. Currently, available immunotherapies show promise for treating metastatic disease in some cancer patients. NK cells were discovered over half a century and are well known for eliminating tumor cells without prior activation; thus, they are in the first line against malignant tumors and metastases. Multiple strategies have been aimed at activating NK function, showing substantial therapeutic effects on metastatic diseases.

### 5.1. Cytokines to Boost Activation of NK Cells

NK cells can be activated by various factors. The cytokines are the first to use the direct engagement of NKR to boost the immune response of NK cells for treating cancer disease. These include IL2, IL12, IL15, IL21, and type I IFNs but not for all [176,177,178,179,180,181,182]. Following cytokine treatment, NK cells convert (turn into) lymphokine-activated killer (LAK) cells, subsequently producing cytokines and upregulating the factors such as adhesion molecules, perforin, granzymes, FasL, and TRAIL [183,184,185,186,187], thereby enhancing their capability to detect and adhere to cancer cells, triggering various activity to eliminate cancer cells through perforin/granzyme-dependent necrosis [188,189] and/or Fasl/TRAIL-mediated apoptosis [184,185,186,187].

**IL2.** Immunotherapeutic strategies against cancer started in the early 1980s. At the time, clinicians used IL2 to activate NK cells by injecting NK cell-stimulating doses of IL2 or preactivated NK cells (LAK cells) for treating patients with primary or metastasized tumors [190]. Although treatment of NK-stimulating doses of IL2 or transplantation of LAK showed some positive effects in patients with advanced cancers [190], IL2 treatment was unfortunately associated with side effects of capillary leak syndrome [191]. Moreover, IL2 enhanced the sensitivity of NK cells to apoptosis when they came into contact with vascular endothelium [192], leading to a reduction in tumor NK cell infiltration.

**IL15.** IL15 could promote NK cell survival and protects NK from activation-induced cell death (AICD). Additionally, it stimulates NK cell expansion more efficiently than IL2 [192,193]. However, relevant high doses of IL15 are required to elicit significant antitumor effects [194].

**IL12.** Unlike IL2 and IL15, IL12 mainly stimulates NK cell-mediated IFNγ production [195,196] to inhibit tumor angiogenesis and stimulate TRAIL/FasL-induced cell apoptosis in various cancers [197,198]. However, IFN-γ stimulated type 1 immunity may counteract the functions of tumor immunosuppressive type 2 cytokines (TGFβ and IL10) [199,200].

**IL21.** IL21 is a promising cytokine with the capability to activate NK cell antitumor immunity [201,202]. It promotes the expression of genes associated with type 1 immune reaction and differentiation of the highly cytotoxic CD56dim/CD16+ NK cells for triggering ADCC against tumor cells [202,203]. However, its treatment appears to have no specificity, and its efficacy was not as expected. Moreover, its treatment does not create cross-resistance or overlapping toxicities as other agents [204]; it still uses in cancer treatment in combination with other cytokine-based immunotherapies [205,206].

**Other cytokines.** The early-acting cytokines such as Flt3L, SCF, and IL7 can also enhance NK cell numbers [207,208]. However, Flt3L did not stimulate a desirable effect for a long time; more studies will be needed to examine the efficacy [209].

**TLR.** The engagement of TLR3 and TLR9 could lead to NK cell activation [108,210]. Due to viral and bacterial products could stimulate immune activity through TLRS, the study has synthesized the molecules from viral and bacterial products to treat the cancer cells in combination with cytokines [211].

Although the promise of cytokines to boost NK cell activation, the side effects of the toxicity of systemic cytokine usages and induced NK cell apoptosis are two significant limitations of cytokine-related NK cell immunotherapies in cancers. To overcome the limitations, a new strategy of dose pulsing of cytokine combinations keeps synergistic antitumor function and manages their toxicities. For example, using low doses of IL2 to promote NK cell proliferation, following high-dose pulses of IL2 or other cytokines to stimulate NK cells [212,213,214]. Additionally, an alternative approach was reported to use fusion proteins to deliver cytokines to tumor cells by tumor-specific Abs in synergy with NK activating cytokines IL21 [214,215].

To efficiently kill metastases, NK cells require the capability to traffic to tumor sites. Chemokines could regulate NK cell migration and stimulate NK cells to infiltrate tumor sites. For example, NK cells express chemokine receptors to respond to CXCL12 and CXC3L1, leading to migration vigorously [19]. Chemokines also regulate the NK cell interaction with other immune cells, such as DC [216,217], which triggers NK cell-associated antitumor immunity [218]. After directly contacting NK cells, DC promotes NK cell survival, activation, and differentiation through the secretion of IL12, IL1, IL18, and IL15 cytokines [219].

### 5.2. Immunomodulatory Agents to Modulate the Activation of NK Cells

Recent studies have renewed interest in using molecules that can block inhibitory pathways of NK cells to counter metastatic disease [220]. For example, one study showed that deletion of the gene-encoded E3 ubiquitin ligase Cbl-b or inactivation of its E3 ligase, allows NK cells to destroy metastatic tumors spontaneously [115]. The TAM tyrosine kinase receptors (Tyro3, Axl, and Mer) were identified to be a ubiquitylation substrate of Cbl-b. Treating mice bearing metastatic melanoma and mammary cancer with a small molecule TAM kinase inhibitor remarkedly reduced metastases and enhanced NK cell-mediated antimetastatic activity [221,222]. Moreover, the anticoagulant warfarin was found to exert antimetastatic activity via Cbl-b/TAM receptors through NK cells, suggesting that the TAM/Cbl-b inhibitory pathway may be the druggable target for awakening the NK cells to kill cancer metastases. Others that can mediate antimetastatic potential through NK cells include CD39/Entpd1 inhibitor polyoxomethalate-1 [166], ADORA2A antagonist PFB-709 [223], immunomodulatory drugs lenalidomide [224] and pomalidomide [225], as well as IDO inhibitors indoximod [226] and epacadostat [227] that inhibit IDO mediated suppression of NK cell function.

### 5.3. Enhancing NK Cell-Mediated Antibody-Dependent Cellular Cytotoxicity

Most matured NK cells express potent activating receptor CD16 that crosslinks with the Fc region of antibodies to initiate ADCC, killing the antibody-coated metastatic cancer cells [228] (Figure 4). Many targeted anticancer antibodies used in the clinic can not only inhibit the targeted signaling pathways for metastatic tumor cell survival and proliferation but also enhance NK cell-induced ADCC [229] (Figure 4), such as FDA-approved cetuximab (antibody for the epidermal growth factor receptor) and trastuzumab (antibody of the erb-b2 receptor tyrosine kinase). The efficacy of these antibodies in treatment depends on NK cells to execute ADCC [230]. For example, studies showed that the therapeutic antibodies for tumor antigens could promote NK cell infiltration to cancer and trigger NK cell-mediated ADCC, suggesting the role of NK cells in adaptive immunity [231]. Another agent ALT-803, a superagonist IL-15 mutant and IL-15Rα-Fc fusion complex, is currently under FDA review, which appears to have the potential to treat patients with advanced solid tumors through activating NK cells and enhancing NK-mediated ADCC [232,233]. For promoting the efficacy of ADCC, substantial progress has been achieved in developing advanced-generation antibodies with ameliorated ADCC potential [234] based on isotype and glycosylation of antibody and NK cell activating receptor CD16 [229]. Furthermore, bispecific and trispecific multifunctional antibodies with Fc recognizing NK cell CD16a and one or more Fc regions for tumor antigens that preserve ADCC potential and physically tie NK cells to malignant cells through bispecific killer engagers (BiKEs) or trispecific killer engagers (TriKEs), therefore ensuing specificity and direct interaction between immune cells and tumors while low systemic toxicity [235,236] (Figure 4). NK cell-engaged BiKEs or TriKEs provide a solution to hurdle the limitations, which are lower cost, less toxicity, and less time-consuming preparation compared to T cell-engaged [63,237]. These agents can create an immunological synapse between NK and tumor cells by simultaneously engaging other NK cell activating receptors (not CD16) with one or two tumor antigens to maximize the lysis of the tumor by NK cells. Several BiKEs and TriKEs have been developed for various targets in preclinical or clinical trials (Table 2) [238,239]. For example, AFM-24, a BiTKe engaged IgG1-scFv fusion antibody of CD16 with epidermal growth factor receptor (EGFR) activating NK cells through CD16 receptor for targeting EGFR, is now in clinical Phase II for treating patients with lung, colorectal cancer, and metastasis [240]. Recently, Vallera and colleagues reported successfully generating a functional TriKE bispecific antibody that could bind CD16 on NK cells, recognize CD33 on myeloid cancer cells, and also contain a modified human IL15 to better induce significant NK cell cytotoxicity and cytokine secretion against CD33+ tumors [241] compared to BiKE only recognize CD16 and CD33 [242]. More recent studies have reported that replacement of “humanized” anti-CD16 single-domain camelid antibody from the anti-CD16 scFv enabled the use of wild-type IL15 instant of modified IL15 to build up cam16-wtIL15–33 TriKE, which exhibited more robust IL15 signaling and NK cell activation leading effective elimination of cancer by NK cells [243]. Thus, this TriKE is a unique NK engager with cytokine signaling, tumor-specific targeting, and a single intramolecular ADCC, which drives specific tumor-killing and induces NK cell proliferation and survival via IL15 fragments.

### 5.4. Blocking (Shielding) the NK Inhibitory Receptor Signals

NK cells undergo functional equilibrium through activating and inhibitory receptors. Metastatic cells often express high levels of ligands for inhibitory receptors to suppress the immune response. For example, NK cells serve as detectors and killers in tumor progression, typically eliminating cells lacking MHC class molecules. However, metastatic cells often manage to express a certain level of downregulated MHC-I and other critical molecules in the antigen presentation pathway, which may inhibit NK cell activation through its inhibitory receptor and limit T cell response to avoid both NK and T cell-mediated elimination [244]. Thus, blockade of the inhibitory signal is essential to evoke the capability of NK cells against metastatic tumors [245]. Specific antibodies against NK cell inhibitory receptors (NKIR) are common agents for blocking NK inhibition signals and inducing activation signals to unleash the capacity of NK cells in antimetastases [246,247]. Several antibodies specific for KIRs, NKG2A, or LIR-1 inhibitory receptors exert substantial therapeutic effects in preclinical models [248,249,250]. For example, in clinical trials, lirilumab targeted KIR2D1/2/3 has been evaluated in multiple patient cohorts with relapsed/refractory myeloma [251,252]. However, clinical trials did not result in the designed efficacy in monotherapy [246] due to the reason of that it might reduce the expression of KIR2D on NK cells and disrupt the NK cell’s “education” process, resulting in a decrease in NK cell quantity and quality [253]. Despite the unexpected in monotherapy, the KIRs antibodies combined with other agents are showing some promise [117,251]. For instance, the combination of lirilumab and lenalidomide was well tolerated and increased the median progression-free survival of multiple myeloma patients [251]. It was also reported that the anti-NKG2A antibody could restore NK cell activation and promote anticancer immunity in combination with PD-1/PD-L1 antibodies for lymphoma [254] as well as the combination with anti-EGFR antibody for treating squamous cell carcinoma [246]. A combination of such anticancer agents could restore NK cell activation by blocking the inhibition of NKG2A and triggering NK cell-mediated ADCC [125]. The TIGIT, CD96, TIM-3, and PD1 inhibitory receptors expressed in NK cells are potential immune checkpoint therapeutic targets [254,255]. Metastatic cancer cells highly express the ligands of these receptors, which leads to NK cell exhaustion and is often associated with unfavorable prognosis. Thus, inhibition of these receptor signaling averts the exhaustion of NK cells and enhances antimetastasis solid effectiveness. In fact, the studies have shown that the combination of anti-TIGIT antibody tiragolumab and anti-PD-L1 antibody atezolizumab exhibited a significant benefit for treating NSCLC patients with high levels of PD-L1 [256,257,258], leading to US FDA to approve the tiragolumab/atezolizumab as the first-line treatment for metastatic NSCLC patients with PD-L1 positive but no EGFR or ALK mutations recently. Other studies using a combination of tiragolumab and atezolizumab for treating solid tumors have also shown significant effects [259]. These indicate that inhibiting PD-1/PD-L1 signaling could significantly promote the capability of NK cells to antitumor progression, implicating that combined therapy with PD-1/PD-L1 on NK cells is a reasonable approach for NK cell-based therapy [260]. Interestingly, recent studies have demonstrated that alloreactive KIR-mismatched NK cells could induce antitumor effects [261,262]. Additionally, the knock of NKG2A inhibitory receptors in NK cells promoted NK cell-induced killing of metastasis [127,263].

### 5.5. Engineered NK Cells (CAR-NK) for Metastasis Therapy

Although many studies demonstrated that NK cells could kill tumor cells in animal models, the clinical benefits of NK cells in cancer therapy are limited. Moreover, conventional immunotherapies have several disadvantages, for example, cytokine-induced toxicity, tumor lysis syndrome, other side effects on normal tissues, and genotoxicity, which may cause death. A new generation of cancer immunotherapy has started to genetically engineer T cells and NK cells that express chimeric antigen receptors (CARs), which have been demonstrated to have promise for treating hematologic cancers [264]. CAR-T or -NK cells express engineered IgG to recognize tumor antigens allowing T cells and NK to destroy the tumor cells precisely and effectively [264,265]. Unlike CAR-T cells, CAR-NK cells are designed to avoid therapy-related toxicity and immune-induced side effects [266,267]. Furthermore, NK cells have significant advantages, including their contribution to the graft-vs-leukemia/graft-vs-tumor effect, and are not responsible for the graft-vs-host disease and multiple various sources [266,267,268,269,270] (Figure 5. Like the structure and design of the CAR-T, the CAR-NK is composed of a promoter, a short signal peptide (SP), a tumor antigen binding domain (scFv with a linker between VH and VL) (ectodomain), a hinge region, a transmembrane domain, and intracellular activation signal domain (endodomain) (Figure 6) [271]. The tumor antigen binding domain derives from single chain variable fragments (scFv) of an antibody that recognizes tumor-specific antigens. The hinge region links a tumor antigen scFV to a transmembrane domain for promoting the dimerization of the CAR, increasing cytokine production and cell proliferation of CAR cells, and enhancing persistence and antitumor effects in vivo. The transmembrane domain leads the CAR structure to anchor on the cell NK cell membrane. Once recognizing tumor cells by the tumor antigen binding domain, the CAR triggers the activation domain signal for killing targeted tumor cells. Numerous CAR-NKs have shown promise in mediating antimetastasis activity in preclinical and clinical trials (Figure 5 and Figure 6) [272,273].

### 5.6. Combination of Multiple Strategies for Metastasis Therapy

Despite the success of CAR-NK cells in some hematologic cancers, their application has been hindered by the high production cost, the time-consuming treatment procedure, and some unexpected side effects. In recent years, a variety of combined approaches have been tested, including NKIR and immune checkpoint blockers, CAR-NK producing Bi-specific scFv and immune checkpoint blocker, CAR-NK with NKIR blocker, CAR-NK with targeted inhibitor, or multifunctional BiKEs and TriKEs (Figure 7) [274,275,276]. The fusion molecules containing one or two NK activating receptors binding scFv or IgG (CD16-binding scFv and NKG2D binding IgG VHH or NKp46 binding Fc) and one or more tumor-specific antigens binding regions have been engineered as the trifunctional or tetrafunctional NK cell engager that targets tumor-specific antigens and activates NK cells for eliminating the tumors significantly with minimized side effects [38,277,278,279]. This approach combined the different strategies for activating NK cells to kill tumors specifically and efficiently, which is particularly innovative.

## 6. Perspectives

Tumor immunology has progressed impressively in developing innovative therapeutic approaches for fighting cancer and metastasis. The remarkable body of research aimed to boost the immune response by activating positive and/or inhibiting negative signals. Antibodies targeting the immune checkpoint have unlatched a new era for the clinical treatment of cancers. The significant research in the mechanism of antibodies has provided awareness to understand how the Fc region of an antibody affects the immune response. Accordingly, BiKEs or TriKEs have increased specificity and facilitated direct interaction between NK cells and tumors with less systemic toxicity. Recently, new techniques can be used to control when and where CAR-NK cells are active, targeting tumors at specific times and preventing CAR-NK cells from becoming less effective over time during cancer treatment. NK cells possess vigorous cytolytic activity in metastatic cancer and have become an attractive tool for antimetastatic immunotherapy. Therefore, more studies on the interactions between metastatic cancer cells and NK cells will be necessary to clarify the mechanism of NK cell cytolytic activity against metastasis and tumor resistance to NK cell-induced destruction. For example, trogocytosis has been observed between CAR cells and tumor cells, causing antigen reduction and tumor relapse [39,280]. A study of the mechanism of trogocytosis in CAR-NK cell therapy will be needed to develop a more effective strategy for CAR-NK application in antimetastatic diseases. Recently, a study using a dual CAR system with an NK self-recognizing inhibitory CAR and an activating CAR against a tumor antigen prevented the trogocytosis-caused tumor resistance to CAR-NK cells mediated elimination, indicating its practical application moving forward [39,281]. In summary, future studies should focus on better understanding the molecular mechanisms of how metastatic tumors escape immune detection and destruction and developing new agents with higher immune specificity, better tolerability, and fewer side effects for clinics to cure metastatic dead diseases successfully.

## Figures and Tables

**Figure 1 cancers-15-02323-f001:**
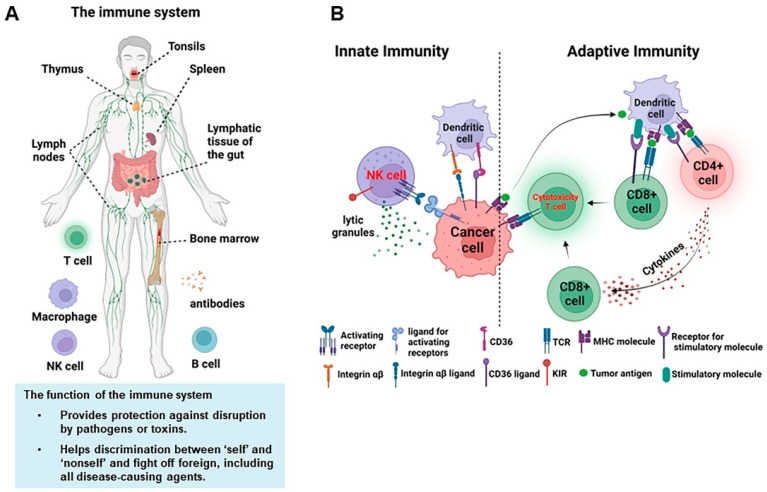
The human immune system and its functions. (**A**) The immune system consists of organs (thymus, tonsils, spleen, lymph, lymphatic tissue of the gut and bone marrow), immune cells (T cells, NK cells, B cells, macrophage, etc.), and molecules (antibodies, cytokines, chemokines, etc.). It protects against foreign pathogens, eliminates cancer cells, and maintains normal body functions. (**B**) Two primary arms of effector immunity are innate and adaptive immunities for immune response. Innate immunity is nonspecific and refers to defense mechanisms activated rapidly to prevent the spread of foreign antigens, including those produced by cancer cells. The main cell types involved in innate immunity are natural killer cells (NK), dendritic cells (DC), and macrophages. NK cells are activated by cancers through NK cell activating receptor interaction with its ligands in cancer cells, releasing perforin and other killer molecules to trigger cancer cell death. Adaptive (acquired) immunity comprises the second line of defense, responding to particular ‘non-self’ antigens expressed in cancer cells. It is characterized by clonal expansion of T and B lymphocytes (T cell and B cell); upon expansion, these clonal cells express the same antigen receptor and are primed to fight the same pathogen. B lymphocytes are primarily involved in antibody-mediated immunity, while T lymphocytes are mainly associated with cell-mediated immunity. After activation through interaction, T cells and tumor antigens represented by DC, CD8+ T cells become cytotoxicity T cells, and CD4+ T cells release cytokines to stimulate CD8+ T cells to convert to cytotoxicity T cells. Cytotoxicity T cells recognize the cancer cells and mediate cancer cell clearance through direct cytotoxicity or secretion of inflammatory cytokines.

**Figure 2 cancers-15-02323-f002:**
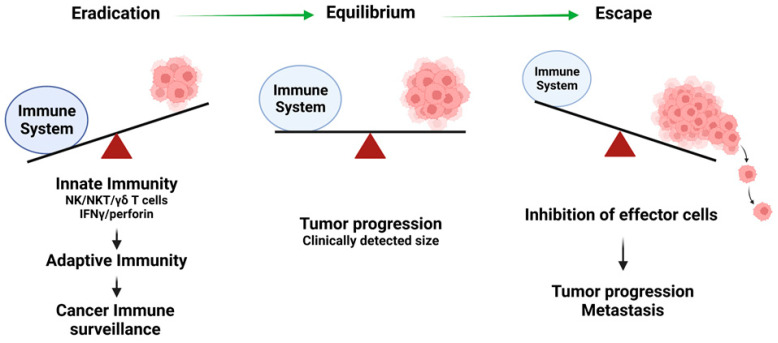
Interactions between the immune system and tumor cells can determine the fate of tumors. Interactions between the immune system and tumor cells are like the relationship between a mouse and a cat game. This process is called cancer immunoediting, which consists of three “Es”: eradication, equilibrium, and escape. In eradication, tumor cells are initially destroyed by the immune system by activating both innate and adaptive mechanisms. However, some tumor cells manage to survive immune destruction and enter what could be a lengthy equilibrium phase. Immunologically sculpted tumors grow progressively during the final escape phase, become clinically mass or metastasize, and establish an immunosuppressive TME.

**Figure 3 cancers-15-02323-f003:**
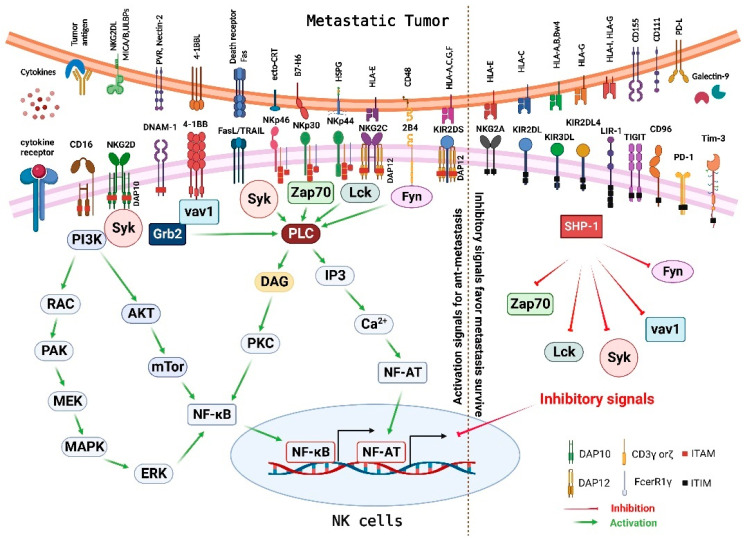
NK cell activity is precisely regulated by a dynamic balance of signals transduced by a diverse set of activating and inhibitory receptors expressed on NK cells. NK cells express an array of activating and inhibitory receptors. Both activating and inhibitory receptor signals jointly mediate the outcome of NK cell target metastatic tumor cells. The activating receptors encounter the ligands expressed in metastatic tumor cells, activate the immunoreceptor tyrosine-based activation motif (ITAM), and activate PI3K and PLC through adaptors such as Syk, Vav1, Zap70, Lck, and Fyn, therefore activating transcriptional factors for stimulating gene expression and promote NK proliferation and release killing molecules for cytotoxicity. As another highly efficient activating receptor, CD16 induces activation signals and triggers antibody-dependent cellular cytotoxicity (ADCC) against antibody-coated cancer cells following binding with the Fc region of antibodies. Moreover, NK cells are activated by cytokines through cytokine receptors. The inhibitory receptors contain the immunoreceptor tyrosine-based inhibition motif (ITIM) in the cytoplasmic tail. After binding to the ligands in metastatic tumor cells, the inhibitory receptors trigger an inhibitory signal through the immunoreceptor tyrosine-based inhibition motif (ITIM), requite the SHP-1 and delivering the inhibition signals through adaptors molecules, therefore blocking gene expression and NK cell proliferation and maintaining NK cell self-tolerance.

**Figure 4 cancers-15-02323-f004:**
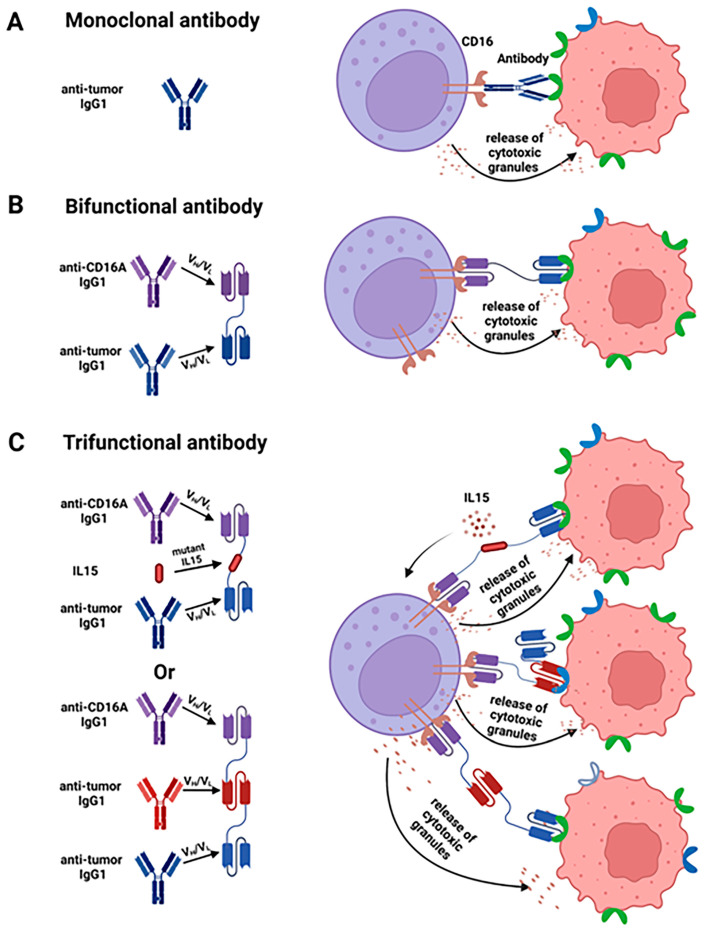
Enhancement of NK cell cytotoxicity (antibody-dependent cellular cytotoxicity, ADCC). (**A**) Matured NK cells express activating receptor CD16 (FcγRIII) that crosslinks with the Fc region of antibodies and initiates antibody-dependent cellular cytotoxicity (ADCC), killing the antibody-coated metastatic cancer cells. (**B**) For promoting the efficacy of ADCC, a novel bispecific IgG1-scFv fusion antibody targeting CD16a on NK cells and specific tumor antigens (bispecific) on metastatic tumor cells has been developed as a second-generation antibody with improved ADCC potential. (**C**) Multifunctional antibodies combined anti-CD16 scFv with IL15 and antimetastatic tumor scFv (bispecific) or anti-CD16 scFv with two antimetastatic tumor scFvs (trispecific) to amplify NK cells and enhance their mediated killing of antigen-expressed metastatic tumor cells. This approach has also been used to physically bridge malignant cells and immune effectors through bispecific killer engagers (BiKEs) or trispecific killer engagers (TriKEs) to increase specificity and facilitate more direct interaction between immune cells and tumors while decreasing systemic toxicity.

**Figure 5 cancers-15-02323-f005:**
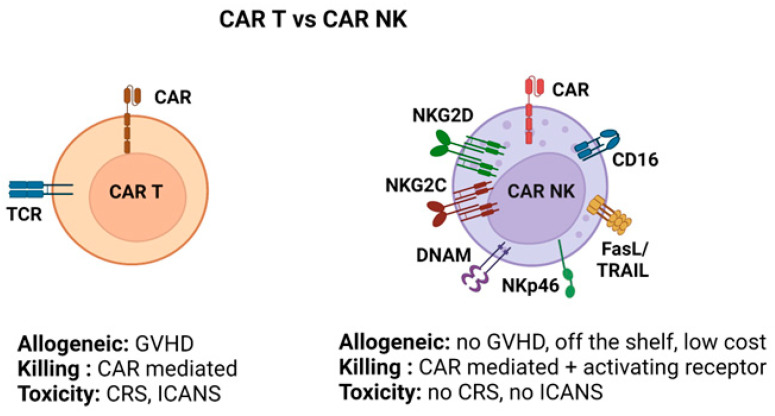
Properties of CAR-NK cells and CAR-T cells. CAR-NK cell-based therapy is superior to CAR-T cell therapy concerning allergenicity, tumor killing, and toxicity. Moreover, CAR-NK cell-based therapy could combine with multiple strategies of various activating receptors.

**Figure 6 cancers-15-02323-f006:**
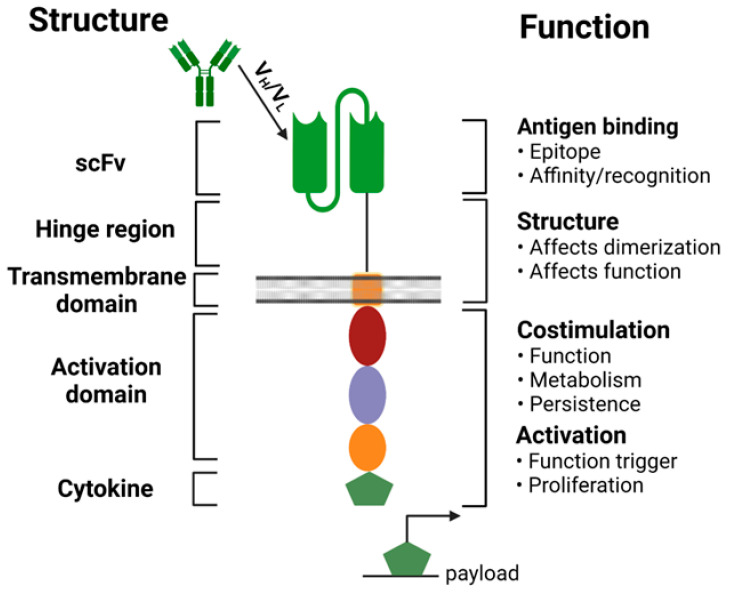
Engineered NK cells (CAR-NK) for metastasis therapy. CARs for NK cells consist of a tumor antigen binding domain (scFv with a linker between VH and VL), a hinge region, a transmembrane domain, and an intracellular activation signal domain. The tumor antigen binding domain contains single chain variable fragments (scFv) derived from antibody recognize tumor-specific antigens. The hinge region links to a transmembrane domain to promote the dimerization of CAR, increase cytokine production and cell proliferation of CAR, and enhance persistence and antitumor effects in vivo. The transmembrane domain allows the CAR structure to anchor on the NK membrane. Recognition of tumor cells by CARs through the tumor antigen binding domain triggers the activation domain signal and leads to tumor cell death. The scFv binds and recognizes the tumor-specific antigen, then activate CAR-NK cells’ intracellular activation domain triggering functional activation and proliferation.

**Figure 7 cancers-15-02323-f007:**
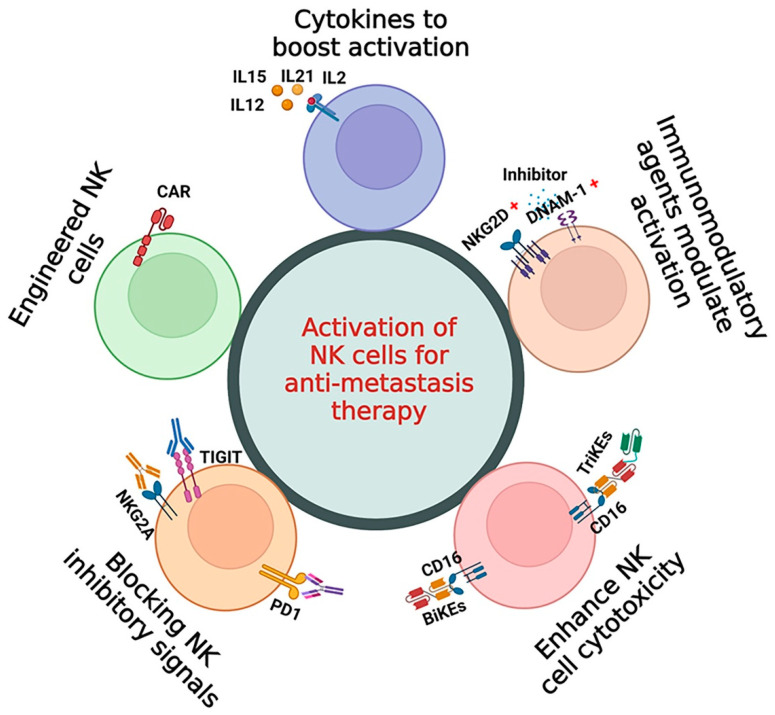
NK cell-based immunotherapy for metastasis therapy. NK cells are known for killing tumor cells without prior stimulation, thus participating in the first line of defense against malignant transformation and metastasis. Multiple strategies aimed at activating NK cell immunosurveillance have demonstrated substantial therapeutic effects in preclinical models of metastatic dissemination. Most therapeutic strategies currently employed in the clinic stem from restoring NK cell activation-dependent immune responses against metastatic cancers.

**Table 1 cancers-15-02323-t001:** Properties of NK cells and T cells in antitumor activity.

Property	NK Cell	T Cells
Type of immune response	Innate immunity, Adaptive Immunity	Adaptive immunity
Cell marker	CD56+CD3-CD16+	CD3+CD4+ or CD3+CD8+
Development	Differentiate in the BM	Differentiate in the thymus
pre-stimulation	No antigen priming	Antigen priming required
Location	Primarily in blood and tissue resident	Antigen specific tumor site
Side effect of overactivation	No/low risk of GVHD	Allogeneic T cells induce GVHD
Mechanism of activation	Recognition of abnormal or missing self-molecules on the surface of cancer cells through the complex array of receptors	Antigen-specific recognition of of cancer cells through TCR
MHC dependence	Do not require MHC matching for antigen recognition and activation	Require MHC matching for antigen recognition and activation
Cytotoxicity	Directly kill cancer cells through the release of cytotoxic granules, greater cytotoxicity	Directly kill cancer cells through the release of cytotoxic granules
Cytokine secretion	Secrete cytokines to stimulate other immune cells to attack cancer cells	Secrete cytokines that stimulate other immune cells to attack cancer cells
Target	Target various cancer cells that have lost MHC expression or express stress-induced molecules	Target cancer cells that express specific antigens on their surface
ADCC	mediated antibody-dependent cellular cytotoxicity of cancer cells	No
Anti-metastatic activity	Highly effective against dissimilating tumor cells and distant site tumor cells	Have limited efficacy against metastatic tumor cells

**Table 2 cancers-15-02323-t002:** Progress of multiple-specific functional (bispecific, trispecific, and tetraspeicific) antibodies in NK cell-based therapy.

Name	Target	Format	Mechanism	Disease	Status
AFM-13	CD30/CD16	scFv-scFv (BiKE)	CD30 inhibitor CD16 regulator	relapsed or refractory MM	phase II
AFM-24	EGFR/CD16	scFv-scFv (BiKE)	EGFR blockers CD16 regulator	advanced solid tumors	phase II
AFM26	BCMA/CD16	scFv-scFv (BiKE)	BCMA blocker CD16 regulator	relapsed or refractory MM	phase I/II
6MW3411	PD-L1/CD16	scFv-scFv (BiKE)	PD-L1 inhibitors CD16 NK cell recruitment agent	solid tumor	pre-clinical
HRS-3/A9 or Anti-CD16/CD30 BiMAB	CD16/CD30	scFv-scFv (BiKE)	CD30 regulator CD16 regulator	hodgkin’s disease	pre-clinical
NKp46 NKCE	NCR1/CD16	scFv-scFv (BiKE)	NCR1 inhibitor CD16 NK cell recruitment agent	tumor	pre-clinical
161533/GTB3550/OXS3550	CD16/IL15/CD33	scFv-IL15-scFv (TriKE)	fusion protein trifunctional, NK cell stimulant, CD16 regulator, CD33 inhibitor	High -risk MDS, relapsed or refractory AML	phase II
cam161533 TriKE	CD16/IL15/CD33	scFv-IL15-scFv (TriKE)	CD33 regulator IL15 regulator CD16 regulator	High -risk MDS, relapsed or refractory AML	pre-clinical
GTB-3650 (humanized CD16scFv)	CD16/IL-15/CD33	scFv-IL15-scFv (TriKE)	fusion protein trifunctional, NK cell stimulant, CD16 regulator, CD33 inhibitor	AML, MDS	pre-clinical
CD16-IL15-CLEC12A	CD16/IL15/CLEC12A	scFv-IL15-scFv (TriKE)	NK cell stimulant ADCC effect	acute myeloid leukemia, Leukemic stem cells	pre-clinical
triplebody	NKG2D/CD19/CD33	ULPB2-scFv-scFv (TriKE)	activating NK cell, CD19 and CD33 inhibitors	mixed lineage leukemia (MLL)	pre-clinical
triplebody	CD33/CD16/CD19	scFv-scFv-scFv (TriKE)	CD16 regulator, CD19 and CD33 inhibitors ADCC	MLL	pre-clinical
sctb	CD123/CD16/CD33	scFv-scFv-scFv (TriKE)	CD16 regulator, CD123 and CD133 inhibitors, ADCC	AML	pre-clinical
SPM-2	CD33/CD16/CD123	scFv-scFv-scFv (TriKE)	CD16 regulator, CD19 and CD133 inhibitors, ADCC	AML	pre-clinical
TriKE	CD16/CD22/CD19	scFv-scFv-scFv (TriKE)	CD16 regulator, CD19 and CD22 inhibitors, ADCC	B-ALL, B-CLL, AML	pre-clinical
ATriFlex	BCMA/CD200/CD16A	scFv-diabody-scFv (TriKE)	CD16 regulator, CD200 and BCMA inhibitors, ADCC	MM	pre-clinical
SAR443579 (ANKET)	CD123/CD16/NKp46	NKp46-Fc-CD123 (TriKE)	fusion protein trifunctional, NK cell stimulant, CD16 regulator, CD123 inhibitor	AML, MDS	pre-clinical
1615EpCAM	CD16/IL-15/EpCAM	scFv-IL15-scFv (TriKE)	fusion protein trifunctional, NK cell stimulant, CD16 regulator, EpCAM inhibitor	Various carcinomas	pre-clinical
1615133	CD16/IL-15/CD133	scFv-IL15-scFv (TriKE)	fusion protein trifunctional, NK cell stimulant, CD16 regulator, CD133 inhibitor	Cancer stem cells	pre-clinical
161519	CD16/IL-15/CD19	scFv-IL15-scFv (TriKE)	fusion protein trifunctional, NK cell stimulant, CD16 regulator, CD19 inhibitor	B-CLL	pre-clinical
cam1615B7H3	CD16/IL-15/B7H3	VHH-IL15-scFv (TriKE)	fusion protein trifunctional, NK cell stimulant, CD16 regulator, B7H3 inhibitor	Ovarian cancer	pre-clinical
cam1615HER2	CD16/IL-15/HER2	VHH-IL15-scFv (TriKE)	fusion protein trifunctional, NK cell stimulant, CD16 regulator, HER2 inhibitor	Ovarian cancer	pre-clinical
cam1615TEM8	CD16/IL-15/TEM8	VHH-IL15-scFv (TriKE)	fusion protein trifunctional, NK cell stimulant, CD16 regulator, TEM8 inhibitor	NSCLC, solid tumor	pre-clinical
TetraKE (TtsAb)	CD16/IL-5/EpCAM/CD133	scFv-IL15-scFv-scFv (tetraKE)	fusion protein trifunctional, NK cell stimulant, CD16 regulator, CD133 and EpCAM inhibitor	CRC	pre-clinical
SEEDbody	EGFR/HER2/NKG2D	IgG-like VHH-based (NKCE)	activating NK cell, EGFR and HER2inhibitors	Breast cancer	pre-clinical
TsAb	EGFR/CD16a/PD-L1	Bs IgG-Fab (NKCE)	EGFR blocker, PD-L1 blocker and CD16 regulator, ADCC	Epidermoid carcinoma	pre-clinical
ANKET	NKp46/CD16/CD19-CD20-EGFR	Fab-Fc-Fab (NKCE)	activating NK cell, CD16 regulator CD19 and CD20 EGFR inhibitors	LNH	pre-clinical
ANKET	NKp46-NKp30/CD16/CD19-CD20	Fab-Fc-Fab(NKCE)	activating NK cell, CD16 regulator CD19 and CD20 inhibitors	B-ALL	pre-clinical
ANKET4	NKp46/CD16/CD20/IL-2v	Fab-Fc-Fab (NKCE)	activating NK cell, CD16 regulator CD20 inhibitors	B-ALL	pre-clinical
HLE-nano-BiKE	CD38/CD16/HSA	VHH-VHH-VHH (nano)	fusion protein trifunctional, NK cell stimulant, CD16 regulator, CD38 inhibitor	multiple myeloma	pre-clinical
DuoBody (DB)-VHH (TtsAb)	HER2/cMET/EGFR-IL6R-NKG2D	bs IgG-VHH-VHH (NKCE)	fusion protein trifunctional, NK cell stimulant, HER2, cMet, EGFR inhibitor	Breast cancer	pre-clinical

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
