# Peer review of "The Function of NK Cells in Tumor Metastasis and NK Cell-Based Immunotherapy"

_cancers, 2023, doi:10.3390/cancers15082323_

Round 1

Reviewer 1 Report

The format of references should be unified. Please see the yellow mark in the attached PDF.

Author Response

I appreciate the reviewer for all the positive comments and suggestions. Our revisions have been made to the manuscript.

Reviewer 2 Report

In this manuscript, Yu gives an overall review on the role of NK cells against cancer and the potential application for these cells in anticancer therapies with a focus on metastatic cancer. This is the most important and novel contribution to the field of this article since similar reviews on NK cells in cancer immunity and immunotherapy have been recently published.

The manuscript is comprehensive, well written and can be of interest for the researchers within the tumor immunobiology and immunotherapy fields. However several inaccuracies throughout the text and figures must be edited.

Fig 1a

Add  tonsils to the immune system depicted.

Fig 1b

This is an incomplete representation of NK-cancer cell interactions. NKG2D is not the only activating receptor involved in cancer recognition and killing. I would rather indicate activating Receptors instead of NKG2D alone. For the  lytic granules indicate also granzymes not only perforin.

Line 53-54: the statement here is too strict. Cells other than NK and DCs are critical for innate immunity such as macrophages.

Table 1 is really general and not entirely correct.  For example NK cells have been described to differentiate in secondary lymphoid tissues (see Caligiuri articles). I would enrich the description  maybe focusing on the anti-tumor  activity of T and NK lymphocytes. The author may also consider to create a different  table to highlight and summarize the anti-tumor acitivity displayed by NK cells vs met cancers, in comparison to NK response vs primary tumors and to T cell responses.

Line 168 LIR-1 can broadly recognize HLA-I  molecules not only HLA-G. This is reported uncorrectly also in figure 3 where other inaccuracies are present . B7-H6 is a ligand for NKp30, but it’s in correspondence of NKp46. For NKp44 some ligands are known and shoule be indicated.  A very recent paper  described a NKp46 ligand (Santara et al Nature 2023). This could be added in the revised version.

Line 231 Please explain what a GEM model is.

Line 251 it should be treating NK cells with activating ..

Check if ref 120 has been appropriately cited. I didn’ t find the information reported in that paper.

Line 297 Explain the main tricks used by met cancer cells

Line 414 Ido inhibitors are cited, however the mechanism of inhibition of IDO has not been mentioned before this point.

In paragraph 5.3 N-803 (formerly ALT-803) should be mentioned

In fig 7 it’s not easy to read upside down…

Author Response

RE: manuscript ID: cancers-2313913

Dear Editors and Reviewers,

Thank you for your correspondence of 8th April 2023 communicating comments from the expert reviewers and editors. We are also pleased with the positive reviews. We have carefully considered these critiques and revised our manuscript. We hope the revised manuscript is now suitable for publication in Cancers.

Specific responses to the points raised by the reviewers are as follows: Our responses are in bold courier new for easier visualization.

Reviewers' comments: 

Reviewer 1

The format of references should be unified. Please see the yellow mark in the attached PDF.

I appreciate the reviewer for all the positive comments and suggestions. Our revisions have been made to the manuscript.

Reviewer 2

In this manuscript, Yu gives an overall review on the role of NK cells against cancer and the potential application for these cells in anticancer therapies with a focus on metastatic cancer. This is the most important and novel contribution to the field of this article since similar reviews on NK cells in cancer immunity and immunotherapy have been recently published.

The manuscript is comprehensive, well written and can be of interest for the researchers within the tumor immunobiology and immunotherapy fields. However several inaccuracies throughout the text and figures must be edited.

I thank the reviewer for all the great comments.

Fig 1a

Add  tonsils to the immune system depicted.

I appreciate the reviewer for the suggestion. Figure 1a has been updated.

Fig 1b

This is an incomplete representation of NK-cancer cell interactions. NKG2D is not the only activating receptor involved in cancer recognition and killing. I would rather indicate activating Receptors instead of NKG2D alone. For the  lytic granules indicate also granzymes not only perforin.

I thank the reviewer for the great suggestions. Fig 1b has been updated.  

Line 53-54: the statement here is too strict. Cells other than NK and DCs are critical for innate immunity such as macrophages.

Thank the reviewer for pointing this out. The revision has been made in the revised manuscript.

Table 1 is really general and not entirely correct.  For example NK cells have been described to differentiate in secondary lymphoid tissues (see Caligiuri articles). I would enrich the description  maybe focusing on the anti-tumor  activity of T and NK lymphocytes. The author may also consider to create a different  table to highlight and summarize the anti-tumor acitivity displayed by NK cells vs met cancers, in comparison to NK response vs primary tumors and to T cell responses.

These are great suggestions. Table 1 has been updated. The reference has been added.   

Line 168 LIR-1 can broadly recognize HLA-I  molecules not only HLA-G. This is reported uncorrectly also in figure 3 where other inaccuracies are present . B7-H6 is a ligand for NKp30, but it’s in correspondence of NKp46. For NKp44 some ligands are known and shoule be indicated.  A very recent paper  described a NKp46 ligand (Santara et al Nature 2023). This could be added in the revised version.

I appreciate the reviewer for pointing these out. Figure 3 has been updated and the reference has been added.   

Line 231 Please explain what a GEM model is.

Sorry to miss the information. GEM is an abbreviation for genetically engineered mouse (GEMmodels.

Line 251 it should be treating NK cells with activating ..

I appreciate the reviewer for pointing these out. The sentence has been corrected.   

Check if ref 120 has been appropriately cited. I didn’ t find the information reported in that paper.

I have checked ref 120 and added more related references.

Line 297 Explain the main tricks used by met cancer cells

I thank the reviewer for the suggestion. The main tricks have been discussed in the revised manuscript.  

 Line 414 Ido inhibitors are cited, however the mechanism of inhibition of IDO has not been mentioned before this point.

I thank the reviewer for pointing this out. The more detailed information has been updated.

In paragraph 5.3 N-803 (formerly ALT-803) should be mentioned

I appreciate the reviewer for this suggestion. The ALT-803 has been mentioned in the revised manuscript.

In fig 7 it’s not easy to read upside down…

Figure 7 has been updated in the revised manuscript.

We hope our revised manuscript is satisfactory to all reviewers and is now acceptable for publication in Cancers.  Thank you for considering our work for your journal.

Sincerely yours,

Yanlin Yu, PhD

Laboratory of Cancer Biology and Genetics

National Cancer Institute, NIH

Building 37, Room 5046

37 Convent Drive, MSC 4264

Bethesda, MD 20892-4264
